# The Influence of Anthropometric Characteristics on Punch Impact

**DOI:** 10.3390/sports13010012

**Published:** 2025-01-08

**Authors:** Manuel Pinto, João Crisóstomo, Gil Silva, Luís Monteiro

**Affiliations:** CIDEFES, Lusófona University, 1749-024 Lisbon, Portugal; jcpcrisostomo@gmail.com (J.C.); gilaires@gmail.com (G.S.); luis.monteiro@ulusofona.pt (L.M.)

**Keywords:** combat sport, punch, anthropometric characteristics, impact force, impact power

## Abstract

Objective: This review examined the influence of anthropometric characteristics, such as body height (BH) and body mass (BM), on the impact of punches in striking-combat sports. Despite their perceived importance for combat strategy, the relationship between these characteristics and punch impact remains unclear. Methods: We included experimental, quasi-experimental and cross-sectional studies. The search was conducted on 30 August 2024, in three databases. The review analyzed 23 studies involving 381 participants (304 men, 30 women, 47 participants of unknown gender). Various instruments were used in the included studies, including ten instruments used to measure impact force and two instruments used to measure impact power. Results: Impact force ranged from 989 ± 116.76 to 5008.6 ± 76.3 N, with rear-hand straight punches and rear-hand hooks producing the greatest force. The PowerKube, a device specifically designed to measure punch impact power, revealed that the rear-hand straight punch generated the highest power, ranging from 15,183.27 ± 4368.90 to 22,014 ± 1336 W. While higher BM categories were associated with stronger punches, BM alone was not the only predictor. Other factors, such as technique, gender, and sport type, also played roles. The relationship between BH and punch impact showed mixed results. Conclusions: The data suggest that while higher BM categories are associated with greater punch impact, BM is not the only determining factor. The relationship between BH and impact also showed mixed results, with no clear association found. The review highlights the lack of a “gold standard” instrument for evaluating punch impact.

## 1. Introduction

Empirically, certain anthropometric characteristics have been noted as crucial for the technical profile of athletes in striking-combat sports, which are characterized by the application of striking techniques that influence their combat strategies. Striking-combat sports aim for temporary incapacitation or the accumulation of points, according to the specific rules of each discipline [1,2].

It has been shown that the choice of punch depends on the distance, position, and movement relative to the target, with body height (BH) and reach being important variables for punch success [3,4]. However, despite these variables being considered important, their relationship with victory remains inconclusive [5,6].

The literature on the technical analysis of striking-combat sports is diverse but inconclusive. Over the past two decades, the technical profiles of athletes and the variables influencing their decisions during movements have been analyzed [3,5]. In this context, most punches in striking-combat sports can be divided into three types: (a) straight punches (front-facing strikes), (b) hooks (lateral movements), and (c) uppercuts (vertical movements) [3].

To increase the likelihood of finishing a fight by incapacitating the opponent, the impact generated by punches has been widely analyzed [7,8,9]. Broadly speaking, the effectiveness of punch impact implicates a complex movement involving both upper and lower body muscles and the proper cooperation of agonist and antagonist muscles [1,10,11,12,13,14,15]. In this regard, Filimonov et al. [11] stated that rear-hand punches can be divided into three main components to generate impact: (a) contribution of arm muscles to the target, (b) trunk rotation, and (c) leg propulsion from the ground. Similarly, Ruddock et al. [16], mentioned three factors that contribute to punch effectiveness: (a) the speed of muscle group activation, (b) arm propulsion, and (c) muscle activation at the moment of impact to effectively transfer displaced mass, termed “stiffening”.

Moreover, evidence shows variations in impact levels between punch types, different body mass (BM) categories, and athletes’ limb lengths [17,18]. In this line, it is important to consider that BM is an import factor, as sports-based categorization divides by BM categories in combat sports to ensure equity in the sports [19,20].

According to the literature, there are three types of punch impact evaluation conducted in combat sports athletes: (a) direct evaluation of relevant inertia, using load cells and platforms directly connected to the target and providing a direct measure of applied impact (e.g., force platforms); (b) indirect evaluation of relevant inertia, involving the indirect measurement of impact by calculating changes in the target’s acceleration caused by impact instead of directly measuring force (e.g., observing bag movement in slow-motion footage); (c) evaluation of impact on the athlete’s limb, such as at the fist, using measuring devices (e.g., load cells in boxing gloves) [4].

Additionally, various studies have measured punch impact using units like force (N), power (W), velocity (m/s), acceleration (m/s^2^), gravitational acceleration (g), and mass (kg) [9,18,21,22,23,24,25,26]. The International System of Units unit for force is the newton (N), and while many studies analyze punch force in newtons, the dynamic nature of punches requires the measurement of impact power in watts (W) [27,28]. Power describes the amount of work performed per unit of time [28]. This concept appears to be crucial to evaluating the intensity and effectiveness of a punch, as it directly relates the force applied and the punch’s execution speed [27].

The aim of this systematic review is to verify the current state of research regarding the analysis of the influence of anthropometric characteristics on punch impact. We hypothesize that anthropometric characteristics, such as BH and BM, influence punch impact. The studies included in this review cover the period from 2000 to 2020.

## 2. Methods

This systematic literature review was conducted following the PRISMA (Preferred Reporting Items for Systematic Reviews and Meta-Analyses) guidelines [29]. The review was registered on the INPLASY website with the registration number: INPLASY202480138 and DOI: 10.37766/inplasy2024.8.0138.

### 2.1. Eligibility Criteria

All studies published to date that report participants’ anthropometric characteristics (e.g., BH, BM) related to the analysis of punch impact in newton (N) or watt (W) units were included. All types of studies were considered for inclusion in the review, except qualitative studies, systematic reviews, and meta-analyses. This review was limited to articles published in English.

### 2.2. Search Strategy

The search was conducted using the electronic databases PubMed, SPORTDiscus, and Web of Science to identify and select relevant studies for inclusion in this review, combining three sets of terms: (i) terms related to the population of interest; (ii) terms related to anthropometric characteristics; and (iii) terms related to punch techniques. The following search terms were used: (boxing OR combat sports OR muay thai OR kickboxing OR karate OR mma OR kung fu) AND (height OR anthropometric characteristics OR body measurements OR anthropometry OR physical attributes) AND (punch OR strength OR impact force OR power OR performance OR activity profile OR punch performance). The search was conducted on 30 August 2024. Additionally, a manual search of the works in the literature cited in the articles and in reference journals was performed.

### 2.3. Study Selection, Data Extraction, and Synthesis

The articles identified in the search were initially deemed potentially eligible based on their titles and abstracts. After a full review and based on the eligibility criteria, the articles for this review were selected. Zotero for Windows was used to manage the references. A specific form was developed for data extraction, including information on the following: i. study characteristics, i.e., authors, year of publication, and design; ii. sample characteristics, i.e., size, gender, age, BH, BM, and body mass index (BMI); iii. type of sport; iv. anthropometric characteristics; v. measurement instruments; and vi. main results.

The qualitative synthesis of the data was presented in a table format. The extracted characteristics and variables are organized by study, in alphabetical order.

### 2.4. Quality Assessment of Studies

To assess the quality of the included studies, an adapted version of the Quality Assessment Tool for Quantitative Studies from the Effective Public Health Practice Project was used [30], as recommended by the Cochrane Public Health Review Group [31] and previously applied by Teixeira et al. [32]. This tool allows for the assessment of experimental and observational studies in eight domains: representativeness (selection bias); study design; confounding factors; blinding; data collection; data analysis; results presentation; and representativeness (exclusions/dropouts). Each domain is classified as strong (good methodological quality), moderate, or weak (low methodological quality), with the final assessment determined according to the evaluations of each domain.

## 3. Results

### 3.1. Description of the Included Studies

The search of the literature performed in the databases PubMed, SPORTDiscus, and Web of Science resulted in 1849 potential studies (Figure 1). Subsequently, 10 studies were added manually. Of the 1859 studies initially identified, 68 were removed as duplicates. A total of 1749 studies were excluded based on title and abstract. The main reasons for exclusion at this stage were that the articles did not report the anthropometric characteristics of the sample or did not present results of interest for the review (e.g., magnitude of punch impact). Additionally, qualitative studies, systematic reviews, and meta-analyses were excluded at this stage. As a result, 42 articles were considered potentially eligible for full-text reading. After reviewing the full texts of the selected articles, 23 were selected that met the inclusion criteria for this literature review.

### 3.2. Characteristics of the Studies

Table 1 and Table 2 detail the characteristics of the studies included in this analysis. The majority of studies (k = 18) had a cross-sectional observational design, three studies had an experimental design, and two were quasi-experimental.

### 3.3. Participant Characteristics

A total of 381 participants (men = 304; women = 30; unknown gender = 47) were included in the 23 studies (Table 1 and Table 2). The average ages of participants within these samples ranged from 17.5 ± 0.5 to 47.5 ± 10.13 years, with average BH ranging from 172 ± 10 to 182 ± 5 cm, and BM ranging from 64.56 ± 12.1 kg to 86.8 ± 17 kg.

### 3.4. Impact Force

Regarding impact force, a total of 262 participants (men = 186; women = 29; unknown gender = 47) were included in the 16 studies (Table 1). The average ages of these samples ranged from 17.5 ± 0.5 to 47.5 ± 10.13 years, BH ranged from 172 ± 10 to 182 ± 5 cm, and BM ranged from 64.56 ± 12.1 kg to 86.8 ± 17 kg.

To simplify the various terminologies presented in the literature, we chose to define the force measurement devices reflected in this review (e.g., dynamometer, load cell, and force sensor) as load cells, as this is the most common definition in the literature and facilitates interpretation, given that all these devices are specific instruments for measuring impact force. Thus, we identified the following 10 different instruments: (1) accelerometer inserted into a dummy’s head and participants’ gloves; (2) tri-axial accelerometer inserted into a dummy’s head; (3) target with accelerometer and load cell inserted; (4) target with a load cell specific to boxing and a force transducer; (5) target on a board with a load cell on the side, transducer, and software; (6) load cell inserted in a wall bag, with transducer and software; (7) load cell inserted in a target, with transducer and software; (8) load cell inserted in a boxing bag, with transducer and software; (9) force platform with a cushioned target; and (10) boxing bag with a load cell and gyroscope transducer inserted. To facilitate the analysis of impact force data, we included studies that presented values in newton (N) units, identifying the highest impact force values by instrument (Figure 2), as well as the type of instrument and main results (Table 1).

### 3.5. Impact Power

Regarding impact power, a total of 119 participants (men = 118; women = 1) were included in the seven studies (Table 2). The average ages of these samples ranged from 24 ± 4 to 29 ± 2 years, average BH ranged from 176.7 ± 6.2 to 181.72 ± 8.28 cm, and BM ranged from 76 ± 7.2 to 80.9 ± 12.24 kg. The studies evaluated only straight punches, and the cross consistently showed higher values than the jab (front hand direct punch).

The studies that analyzed power used only two instruments, the PowerKube and the StrikeMate; these are similar, with minor differences (name and appearance). The device in question is a portable and lightweight impact cube specifically designed to measure and analyze the impact power of punches. Inside the cube are two high-precision accelerometers responsible for detecting and quantifying the acceleration caused by impacts [33].

### 3.6. Results Synthesis

All studies (k = 23) mention at least one of the participants’ anthropometric characteristics (Table 1 and Table 2).

Regarding impact force, the instruments most commonly used in the studies were those with load cells inserted in the wall bag, equipped with transducers and appropriate software (k = 3), and force platforms with cushioned targets (k = 3). Both instruments, which are direct methods of relevant inertia assessment, showed the punches with the highest values, rear-hand cross and hook, as having an average impact force ranging from 1331.67 ± 234.49 to 5008.6 ± 76.3 N. The highest impact force levels per instrument ranged from 989 ± 116.76 to 5008.6 ± 76.3 N, with the rear-hand straight punch (cross) and rear-hand hook identified as generating the highest impact force levels (Table 1). The instrument that demonstrated the highest impact force values was the one with load cells inserted in the wall bag (Figure 2).

Regarding impact power, all studies used the same type of instrument for direct relevant inertia assessment. This instrument identified the cross as the punch with the highest value, with an average impact power ranging from 15,183.27 ± 4368.90 to 22,014 ± 1336 W (Table 2).

### 3.7. Methodological Quality of the Studies

Table 3 and Table 4 present the methodological quality of the studies. Most studies (k = 21) were rated as having “moderate” methodological quality due to the study design, low sample power, and lack of representativeness of the population studied. Two studies were rated as having “strong” methodological quality.

## 4. Discussion

Striking-combat sports are highly complex and aim to deliver effective strikes while minimizing exposure to counterattacks [22,34]. Athletes perform various punches that can be influenced by individual characteristics such as BM, BH, limb length, and muscle mass, each of which appears to play a crucial role in determining punch impact [17,25,26].

Our literature review specifically focused on the influence of anthropometric factors on punch impact. The data suggest that while higher BM categories generally correspond to greater punch impact, BM alone does not determine punch impact. We emphasize that our findings are descriptive, and direct correlations cannot be inferred due to the variability in measurement methodologies and the use of different instruments across studies. Other factors, such as technique, strength, limb length, gender, and the type of combat sport, are also important. For example, studies comparing athletes of different BM show that athletes with higher average BM tend to generate higher impact forces, but differences in punch impact are not always straightforward [14,44]. Upon examining the studies where BM was analyzed, we found that athletes with higher BM produced greater impact forces. In this regard, Loturco et al. [14] reported lower force levels (1331.67 ± 234.49 N) for participants with an average BM of 64.56 ± 12.1 kg, while de Souza and Marques [45] observed higher impact force levels (2260.79 ± 538.44 N) in participants with an average BM of 86.8 ± 17 kg. However, the study by Loturco et al. [14] included participants of both genders, which may have introduced bias, as men typically produce higher punch impact forces. Additionally, differences in the measurement devices (e.g., calibration, sensor placement, and data processing) used in these studies may account for some of the variability observed. Studies involving only male participants with similar BM (67.7 kg and 67 ± 10 kg) reported higher impact forces than the study by de Souza and Marques [45], indicating that while BM influences impact force, it is not the only determining factor [8,9,45].

Contrary to the assumption that athletes with higher BM consistently produce punches that generate more impact, our review revealed mixed findings regarding BM and impact power. For instance, participants with lower BM (76.5 ± 10 kg) exhibited higher impact power (22,014 ± 1336 W) compared to those with higher BM (80.9 ± 12.2 kg), who showed lower impact power (15,431 ± 4294 W) (Table 3). This variability could also reflect differences in variations in data collection protocols used to measure impact power. Additionally, this variation may be attributed to differences in the types of combat sports, as participants with higher BM were not specified as specialists in punching techniques (e.g., boxing practitioners), which could explain the lower impact power [33,50]. When exploring the relationship between BH and punch impact, no clear causal relationship was found. While a group of participants with an average BH of 178 ± 4 cm demonstrated the highest impact power (22,014 ± 1336 W), those with a slightly taller (181.7 ± 8.3 cm) and shorter (176.7 ± 6.2 cm) BH exhibited lower impact powers (19,640 ± 1410 W and 15,227 ± 225 W, respectively). The small BH differences among participants may have limited the ability to detect a consistent relationship between BH and punch impact power.

It is also worth noting that previous research has attributed importance to the significance of BH, not only as a determinant of athletic performance but also as a potential factor in competition category divisions. For instance, it has been proposed that incorporating BH alongside BM in classification systems could mitigate the prevalence of health problems among athletes while providing competitive advantages in striking-combat sports [19,20].

Additionally, variations in measurement instruments and techniques used across studies might have influenced the results. The lack of significant differences and the variability of results may reflect confounding variables such as skill level, gender, or sport specialization, which were not systematically analyzed.

## 5. Limitations

The aim of this review was to reflect on what the literature mentions regarding anthropometric characteristics and punch impact; however, relevant studies with information on each variable may have been excluded for not meeting the inclusion criteria. Additionally, the use of mean values and comparisons across studies limits the ability to establish causal relationships or direct correlations. Other main reasons for this limitation include the lack of presentation of the participants’ anthropometric characteristics and/or the use of different measurement units to analyze punch impact. According to the literature, several studies measured punch impact using units measuring force (N), power (W), velocity (m/s), acceleration (m/s^2^), gravitational acceleration (g), and mass (kg). These units were evaluated with different instruments, such as accelerometers, high-speed cameras, force platforms, load cells, and transducers. This diversity of equipment and measurements complicates standardization and analysis of results, potentially causing bias. Thus, only studies with measurements in newtons (N) and watts (W) were included.

Another limitation was the homogeneity of the samples. Most studies included male participants, and the influence of gender or other confounding factors on punch impact was not systematically analyzed. This lack of diversity may have compromised the ability to detect significant differences in punch impact among athletes with different anthropometric characteristics.

## 6. Conclusions

Although BM and BH may influence punch impact, the relationship between these factors is complex and multifaceted. Our review highlights the multifactorial nature of punch impact, in which anthropometric characteristics interact with technical skill, training, and sport-specific demands. While athletes with higher BM often produce greater impact forces, exceptions occur, emphasizing the need for further investigation. Additionally, more focused studies on how anthropometric characteristics interact with technical skills and individual physical capabilities could provide a deeper understanding of punch performance, ultimately leading to more targeted training strategies in striking-combat sports.

## 7. Future Implications

Future research should include more diverse samples in terms of their anthropometric characteristics to better elucidate how these variables influence punch impact. The creation and use of a gold standard instrument would facilitate the standardization of analysis methods for measuring punch impact, which is crucial for advancing research in this area, allowing for the comparison of results between different studies and the identification of more consistent patterns among distinct samples. Additionally, systematic exploration of confounding variables such as gender, sport-specific training, and technique is necessary to understand their relative contribution to punch performance. Studies aimed at analyzing the influence of anthropometric characteristics and the strength/power of upper and lower limbs on the performance of different punching techniques will contribute to a deeper understanding of the determining factors in the generation of impact in these movements.

## 8. Practical Application

From a practical perspective, coaches and athletes should be mindful that BM alone may not be the best predictor of punch impact. While athletes with higher BM may generate greater impact forces in certain cases, factors such as combat sport type, technique, strength, and limb length are equally important. Coaches should consider tailoring training programs to improve technique and strength for athletes, regardless of their BM, to maximize punching efficiency. Additionally, athletes specializing in boxing techniques may produce different impact forces compared to those who participate in other combat sports, which suggests that training specificity plays a significant role in the effectiveness of punches.

## Figures and Tables

**Figure 1 sports-13-00012-f001:**
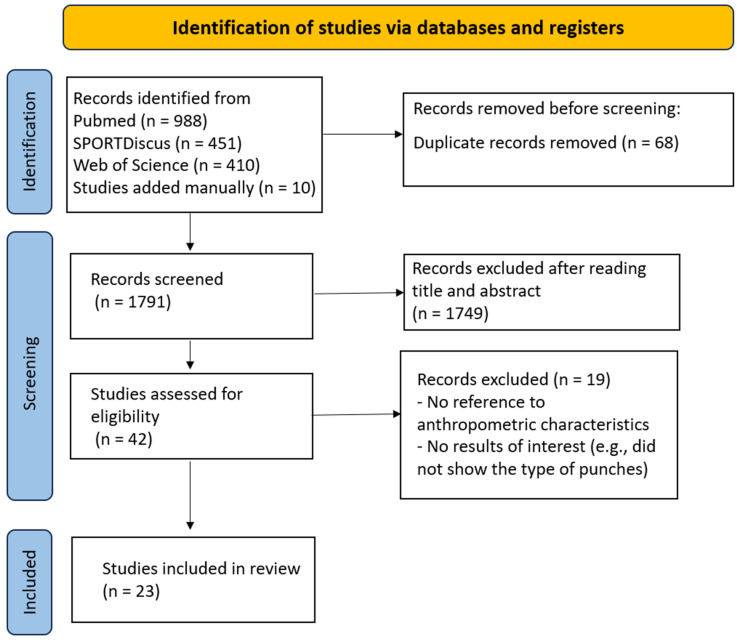
Flowchart according to the PRISMA 2020 guidelines.

**Figure 2 sports-13-00012-f002:**
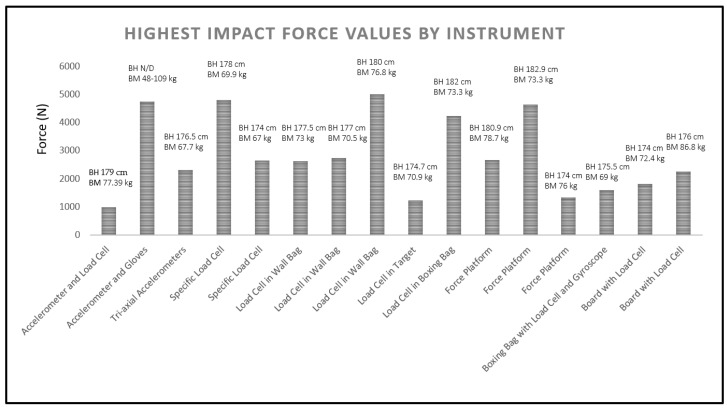
Highest impact force values by instrument. Note: Accelerometer and Load Cell = Target with an accelerometer and load cell inserted; Accelerometer and Gloves = Accelerometer inserted in the dummy’s head and the participants’ gloves; Tri-axial Accelerometers = Tri-axial accelerometers inserted in the dummy’s head; Specific Load Cell = Target with a load cell specific to boxing and force transducer; Load Cell in Wall Bag = Load cell inserted in the wall bag, with transducer and respective software; Load Cell in Target = Load cell inserted in the target, with transducer and respective software; Load Cell in Boxing Bag = Load cell inserted in the boxing bag, with transducer and respective software; Force Platform = Force platform with cushioned target; Boxing Bag with Load Cell and Gyroscope = Boxing bag with load cell and an inserted gyroscope transducer; Board with Load Cell = Target on a board with a lateral load cell, transducer, and respective software. BH: body height; BM: body mass; N/D: no description.

**Table 1 sports-13-00012-t001:** Study characteristics: participants, types of sport, anthropometric characteristics, instruments, and mean and maximum impact force in newton (N) units of measurement.

Reference	Study Design	Participants	Type of Sport	Anthropometric Characteristics	Instruments	Main Results
Adamec et al., 2021 [32]	Cross-sectional	N = 50 (men = 29; women = 21); Age: 34 years; BH: 174 cm; BM: 76 kg.	Karate.	BH, BM.	Force platform with cushioned target (DIRA).	Fmax—Straight punch (4639 N).
Buśko et al., 2016 [33]	Cross-sectional	N = 13 men; Age: 17.5 years; BH: 175.5 cm; BM: 69 kg.	Boxing.	BH, BM.	Boxing bag with load cell and an inserted gyroscope transducer (DIRA).	Fmean—Cross (1592.5 ± 507.1 N).
Chadli et al., 2014 [34]	Cross-sectional	N = 11 (unknown gender); Age: 23.5 ± 0.5 years; BH: 179 ± 9 cm; BM: 77.39 ± 11 kg.	Boxing.	BH, BM.	Target with an accelerometer and load cell inserted (DIRA).	Fmean—Punch (989 ± 116.76 N).
Dunn et al., 2019 [35]	Cross-sectional	N = 15 men; Age: 17.5 ± 0.5 years; BH: 177.5 ± 9 cm; BM: 73 ± 14 kg.	Boxing.	BH, BM.	Load cell inserted in the wall bag, with transducer and respective software (DIRA).	Fmean—Jabs (841 ± 180 N); Cross (1818 ± 332 N); Hooks (2622 ± 288 N).
Dunn et al., 2022 [36]	Cross-sectional	N = 28 men; Age: 19 ± 2 years; BH: 177 ± 7.3 cm; BM: 70.5 ± 11 kg.	Boxing.	BH, BM.	Load cell inserted in the wall bag, with transducer and respective software (DIRA).	Fmean—Jab (823 ± 271 N); Cross (1830 ± 387 N); Lead-hand hook (2491 ± 492 N); Rear-hand hook (2742 ± 571 N).
Dyson et al., 2005 [37]	Cross-sectional	N = 6 men; Age: 24.5 ± 3.3 years; BH: 182 ± 5 cm; BM: 73.3 ± 19 kg.	Boxing.	BH, BM.	Load cell inserted in the boxing bag, with transducer and respective software (DIRA).	Fmean—Cross (4236 ± 181 N); Jab (2722 ± 75 N).
Finlay, 2022 [38]	Experimental	N = 10 men; Age: 19.7 ± 1.2 years; BH: 180.9 ± 7.0 cm; BM: 78.7 ± 9.6 kg.	Boxing.	BH, BM.	Force platform with cushioned target (DIRA).	Fmax—Rear-hand hook (2673 N); Lead-hand hook (2565 N); Cross (2538 N).
Kim et al., 2018 [8]	Quasi-experimental	N = 15 men; Age: 23.4 years; BH: 176.5 cm; BM: 67.7 kg.	Boxing.	BH, BM.	Tri-axial accelerometers inserted in the dummy’s head (DIRA).	Fmax—Cross (2313 N).
Lee and McGill, 2017 [39]	Quasi-experimental	N = 12 men; Age: 24.2 ± 2.9 years; BH: 180 ± 5 cm; BM: 76.8 ± 9.7 kg.	Muay Thai.	BH, BM.	Load cell inserted in the wall bag, with transducer and respective software (DIRA).	Fmean—Jab (3093.7 ± 69.4 N); Cross (5008.6 ± 76.3 N); Knee (9482 ± 152.8 N).
Loturco et al., 2016 [14]	Cross-sectional	N = 15 (men = 9; women = 6); Age: 25.9 ± 4.7 years; BH: 172 ± 10 cm; BM: 64.56 ± 12.1 kg.	Boxing.	BH, BM.	Force platform with cushioned target (DIRA).	Fmean—Jab/men (1152.22 ± 246.87 N); Cross/men (1331.67 ± 234.49 N); Jab/women (902.50 ± 213.49 N); Cross/women (994.17 ± 221.14 N).
Neto et al., 2009 [40]	Cross-sectional	N = 12 (men = 10; women = 2); Age: 23.4 years; BH: 174.7 ± 4 cm; BM: 70.9 ± 12 kg.	Kung Fu.	BH, BM.	Load cell inserted in the target, with transducer and respective software (DIRA).	Fmax—Punch (1226 N).
Smith et al., 2000 [41]	Cross-sectional	N = 23 men; Age: 23.1 ± 1.2 years; BH: 178 ± 6 cm; BM: 69.9 ± 8.6 kg.	Boxing.	BH, BM.	Target with a load cell specific to boxing and force transducer (DIRA).	Fmean—Cross: elite (4800 ± 227 N), intermediate (3722 ± 133 N), beginner (2381 ± 116 N).
Smith, 2006 [9]	Cross-sectional	N = 29 (unknown gender); Age: 21 ± 2 years; BH: 174 ± 8 cm; BM: 67 ± 10 kg.	Boxing.	BH, BM.	Target with a load cell specific to boxing and force transducer (DIRA).	Fmean—Jab to the face (1722 ± 700 N), to the body (1682 ± 636 N); Cross to the face (2643 ± 1273 N), to the body (2646 ± 1083 N); Lead-hand hook to the face (2412 ± 813 N), to the body (2414 ± 718 N); Rear-hand hook to the face (2588 ± 1040 N), to the body (2555 ± 926 N).
V. A. de Souza and Marques, 2017 [42]	Cross-sectional	N = 8 men; Age: 20.25 ± 4.13 years; BH: 174 ± 4 cm; BM: 72.4 ± 9.6 kg.	Karate.	BH, BM.	Target on a board with a lateral load cell, transducer, and respective software (DIRA).	Fmax—Straight punch (1812.01 N).
V. de Souza and Marques, 2017 [43]	Cross-sectional	N = 8 men; Age: 47.5 ± 10.13 years; BH: 176 ± 3 cm; BM: 86.8 ± 17 kg.	Karate.	BH, BM.	Target on a board with a lateral load cell, transducer, and respective software (DIRA).	Fmean—Straight punch (2260.79 ± 538.44 N).
Walilko et al., 2005 [24]	Cross-sectional	N = 7 (unknown gender); BM: 48–109 kg.	Boxing.	BM.	Accelerometer inserted in the dummy’s head and the participants’ gloves (DIRA and AL).	Fmax—Straight punch (4741 N). BM categories: Flyweight (3914 N); Light welterweight (3621 N); Middleweight (3072 N); Super heavyweight (4741 N).

DIRA = Direct inertia-relevant assessment; AL = Assessment on the athlete’s limb; BH: Body height; BM: Body mass; Jab = Straight punch with the lead hand; Cross = Straight punch with the rear hand; Fmax = Maximum punch force; Fmean = Mean punch force.

**Table 2 sports-13-00012-t002:** Study characteristics: participants, types of sport, anthropometric characteristics, instruments, mean and maximum impact power in watts (W) units of measurement.

Reference	Study Design	Participants	Type of Sport	Anthropometric Characteristics	Instruments	Main Results
Brown et al., 2020 [44]	Cross-sectional	N = 15 men; Age: 24.2 ± 2.9 years; BH: 176.7 ± 6.2 cm; BM: 79.3 ± 11.8 kg; BMI: 24.9 kg·m^−2^.	Boxing	BH, BM.	PowerKube (DIRA).	Pmean—Cross (15,227.4 ± 225 W).
Brown et al., 2021 [45]	Experimental	N = 20 men; Age: 28 ± 6 years; BH: 178 ± 4 cm; BM: 76.5 ± 10 kg.	Boxing.	BH, BM.	PowerKube (DIRA).	Pmean—Cross (22,014 ± 1336 W).
Brown et al., 2022 [21]	Cross-sectional	N = 22 men; Age: 28 ± 2 years; BH: 178 ± 8.1 cm; BM: 79 ± 7.1 kg; BMI: 24.9 ± 2.5 kg·m^−2^.	Boxing	BH, BM.	PowerKube (DIRA).	Pmean—Cross (15,227 ± 2250 W).
Brown et al., 2023 [46]	Cross-sectional	N = 16 (men = 15; women = 1); Age: 24 ± 4 years; BH: 181.72 ± 8.28 cm; BM: 80.16 ± 11.32 kg.	Boxing; Muay Thai	BH, BM.	PowerKube (DIRA).	Pmean—Cross (19,640 ± 1410 W).
Del Vecchio et al., 2018 [47]	Experimental	N = 17 men (10 EG; 6 CG); Age: 28 ± 2 (EG) e 29 ± 2 (CG) years; BH: 178 ± 8.1 (EG) e 177.7 ± 5.7 (CG) cm; BM: 79 ± 7.1 (EG) e 79.8 ± 11.9 (CG) kg.	Combat sports (not specified)	BH, BM.	StrikeMate (DIRA).	Pmean—Jab (6781.6 ± 2178.9 W); Cross (15,335.9 ± 4432.8 W); Front kick (8357.5 ± 2895.9 W); Roundhouse kick (40,129.2 ± 10,169.8 W).
Del Vecchio et al., 2019 [48]	Experimental	N = 16 men (10 EG; 6 CG); Age: 25.2 ± 1.8 (EG) e 29 ± 2 (CG) years; BH: 178.1 ± 7.1(EG) e 177.7 ± 5.7 (CG) cm; BM: 76 ± 7.2 (EG) e 79.8 ± 11.9 (CG) kg.	Combat sports (not specified)	BH, BM.	StrikeMate (DIRA).	Pmean—Jab (7478.82 ± 2994.36 W); Cross (15,183.27 ± 4368.90 W); Front kick (7438.64 ± 1910.56 W); Roundhouse kick (45,278.30 ± 11,323.13 W).
Del Vecchio et al., 2021 [49]	Cross-sectional	N = 13 men; Age: 28.8 ± 4.57 years; BH: 176.9 ± 4.14 cm; BM: 80.9 ± 12.24 kg; BMI: 25.9 ± 3.8 kg·m^−2^.	Combat sports (not specified)	BH, BM.	StrikeMate (DIRA).	Pmean—Jab (8081 ± 3742 W); Cross (15,431 ± 4294 W); Front kick (8563 ± 3095 W); Roundhouse kick (46,377 ± 12,209 W).

BH: Body height; BM: Body mass; BMI: Body mass index; DIRA = Direct inertia-relevant assessment; EG = Experimental group; CG = Control group; Jab = Straight punch with the lead hand; Cross = Straight punch with the rear hand; Pmean = Mean punch power.

**Table 3 sports-13-00012-t003:** Assessment of the methodological quality of studies on the impact force of punches.

Reference	Study Design	Blinding	Representativity (Selection Bias)	Sample Representativity (Dropouts)	Confounding Factors	Data Selection	Data Analysis	Representation of Results	Overall Classification
Adamec et al., 2021 [32]	2	3	3	4	2	1	1	1	Moderate
Buśko et al., 2016 [33]	2	3	3	4	2	1	1	1	Moderate
Chadli et al., 2014 [34]	2	3	3	4	2	1	1	1	Moderate
Dunn et al., 2019 [35]	2	3	2	4	2	1	1	1	Moderate
Dunn et al., 2022 [36]	2	3	2	4	2	1	1	1	Moderate
Dyson et al., 2005 [37]	2	3	2	4	2	1	1	1	Moderate
Finlay, 2022 [38]	1	3	3	1	1	1	1	1	Strong
Kim et al., 2018 [8]	1	3	3	1	2	1	1	1	Moderate
Lee and McGill, 2017 [39]	1	3	3	1	1	1	1	1	Strong
Loturco et al., 2016 [14]	2	3	2	4	2	1	1	1	Moderate
Neto et al., 2009 [40]	2	3	2	4	2	1	1	1	Moderate
Smith et al., 2000 [41]	2	3	1	4	2	1	1	1	Moderate
Smith, 2006 [9]	2	3	1	4	3	1	1	1	Moderate
V. A. de Souza and Marques, 2017 [42]	2	3	3	4	2	1	1	1	Moderate
V. de Souza and Marques, 2017 [43]	2	3	3	4	2	1	1	1	Moderate
Walilko et al., 2005 [24]	2	3	3	4	3	1	1	1	Moderate

Legend: 1 = Strong; 2 = Moderate; 3 = Weak; 4 = No rating.

**Table 4 sports-13-00012-t004:** Assessment of the methodological quality of studies on the impact power of punches.

Reference	Study Design	Blinding	Representativity (Selection Bias)	Sample Representativity (Dropouts)	Confounding Factors	Data Selection	Data Analysis	Representation of Results	Overall Classification
Brown et al., 2020 [44]	2	3	3	4	2	1	1	1	Moderate
Brown et al., 2021 [45]	2	3	3	4	2	1	1	1	Moderate
Brown et al., 2022 [21]	2	3	3	4	2	1	1	1	Moderate
Brown et al., 2023 [46]	2	3	3	4	2	1	1	1	Moderate
Del Vecchio et al., 2018 [47]	1	2	3	1	2	1	1	2	Moderate
Del Vecchio et al., 2019 [48]	1	2	3	1	2	1	1	2	Moderate
Del Vecchio et al., 2021 [49]	2	3	3	4	2	1	1	1	Moderate

Legend: 1 = Strong; 2 = Moderate; 3 = Weak; 4 = No rating.

## Data Availability

All original data are freely available in the electronic databases.

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
