# Peer review of "The Influence of Anthropometric Characteristics on Punch Impact"

_sports, 2025, doi:10.3390/sports13010012_

Round 1
Reviewer 1 Report
Comments and Suggestions for Authors
Congratulations on the great effort you put into preparing this article. Please refer to the comments in the review attached.
best regards

Author Response
Thank you very much for taking the time to review this manuscript. Please find the detailed responses below.
Comments 1: Where the phrase height is used, please change it to body height, this applies to both the manuscript and the tables. Please correct.
Response 1: We have revised the manuscript and all tables to replace the term "height" with "body height" as requested. This correction has been applied consistently throughout the document to ensure clarity and precision.
Comments 2: Tables 3 and 4 are incorrectly formatted in the reference line, please correct.
Response 2: We have corrected the formatting issues in the reference lines of Tables 3 and 4. The updated tables now adhere to the required formatting standards.
Comments 3: I have doubts about the analyses on different measuring devices, which may be the differences in the measurement results, please explain.
Response 3: We understand your concerns about the differences in measurement results across various devices. To address this, we have expanded the discussion to include a detailed explanation of how device-specific factors (e.g., sensor placement, calibration, and data processing algorithms) may contribute to variability in impact force values. We also acknowledge the limitations of comparing studies that used different instruments and have added this to the limitations section. Additionally, we have revised Figure 2 to provide more context by incorporating anthropometric data (body height and body mass), helping readers interpret the variability in measurements.
Comments 4: The discussion is a bit modest, please expand.( maybe in terms of differences in striking techniques in the mentioned combat sports, please consider this).
Response 4: The discussion has been expanded to address differences in striking techniques across the combat sports mentioned. We have highlighted how factors such as punching trajectory, force application, and technical execution may influence the impact forces recorded. This addition provides a more comprehensive analysis of the findings and their relevance to different combat sports.
Comments 5: Please add practical implications resulting from the work from the perspective of coaches and athletes
Response 5: We have added a section on the practical implications of our findings. This section outlines how coaches and athletes can use the insights from this study to optimize training strategies, such as tailoring training to specific anthropometric profiles to maximize impact force or selecting equipment that aligns with their performance goals. These recommendations aim to bridge the gap between research and practice.
We appreciate the constructive feedback, and we believe these revisions have significantly strengthened the manuscript. If there are any further suggestions or clarifications needed, we are happy to address them.
Reviewer 2 Report
Comments and Suggestions for Authors
Dear Authors,
Thank you for the opportunity to review your manuscript.
The compilation of the studies you have presented is a helpful process and the work you have done is commendable. Nevertheless, I am of the opinion that you cannot answer the question you have phrased in this way. It is not clear to me how the conclusion that striking power does not depend on anthropometric data can be derived from the available studies.
Your manuscript is more of a summary review; in Tables 1 + 2, the mean values of height, weight and strength/power are presented. However, neither you nor the reported studies performed a correlation calculation that would answer the question of the study.
To answer the phrased hypothesis, the original data of each study would have to be correlated with the respective measured strength values. Currently, you compare the mean values of the individual studies and conclude from this that there are no correlations between weight/height and impact strength. I consider this to be problematic because, as you yourself point out, the study designs are inhomogeneous and the measured strength/power values were collected differently. Likewise – and this is a crucial point – other possible confounding variables were not or only partially recorded in the studies. It is therefore not clear how the claim “Other factors, such as technique, gender, and sport type, also played roles” can be derived from the studies you evaluated.
Your study should have included a crucial analysis, namely plotting the punch impacts achieved against the weight of the athletes. However, this would also have the limitation that only mean values would be plotted against each other, so that correlations cannot be determined with certainty.
Accordingly, Figure 2 is not very meaningful either – it merely shows the variety of measurement methods that have led to completely different values.
If a meta-analysis is not possible, at least a cross-study evaluation should be carried out and the results interpreted cautiously.
Author Response
Thank you for your detailed and constructive feedback on our manuscript. We appreciate the opportunity to address your comments and to improve the clarity and robustness of our work. Below, we provide a point-by-point response to the issues raised:
Comments 1: The compilation of the studies you have presented is a helpful process and the work you have done is commendable. Nevertheless, I am of the opinion that you cannot answer the question you have phrased in this way. It is not clear to me how the conclusion that striking power does not depend on anthropometric data can be derived from the available studies.
Response 1: We acknowledge that our manuscript presents findings from studies that used varied methodologies, making direct correlations between anthropometric data and punch impact challenging. The conclusion that striking power/strength does not depend solely on anthropometric data was intended to highlight the multifactorial nature of punch impact, including factors such as technique, sport type, and gender. However, we recognize that without conducting a direct correlation analysis, this claim may lack sufficient support. We will revise the manuscript to explicitly state this limitation and avoid overgeneralizations.
Comments 2: Your manuscript is more of a summary review; in Tables 1 + 2, the mean values of height, weight and strength/power are presented. However, neither you nor the reported studies performed a correlation calculation that would answer the question of the study.
Response 2: You correctly pointed out that comparing mean values across studies does not establish correlations. We will clarify in the manuscript that the presented mean values are descriptive and do not infer direct relationships. Additionally, we will emphasize that the inhomogeneity of study designs and measurement techniques limits the generalizability of findings.
Comments 3: To answer the phrased hypothesis, the original data of each study would have to be correlated with the respective measured strength values. Currently, you compare the mean values of the individual studies and conclude from this that there are no correlations between weight/height and impact strength. I consider this to be problematic because, as you yourself point out, the study designs are inhomogeneous and the measured strength/power values were collected differently. Likewise – and this is a crucial point – other possible confounding variables were not or only partially recorded in the studies. It is therefore not clear how the claim “Other factors, such as technique, gender, and sport type, also played roles” can be derived from the studies you evaluated.
Response 3: We agree that confounding variables, such as technique and gender, were not systematically analyzed across the included studies. The statement regarding their influence was based on study discussions and observed variability. To address this, we will include a paragraph that explicitly discusses the limitations of the existing literature in capturing these variables and how they may impact punch impact.
Comments 4: Your study should have included a crucial analysis, namely plotting the punch impacts achieved against the weight of the athletes. However, this would also have the limitation that only mean values would be plotted against each other, so that correlations cannot be determined with certainty.
Response 4: Conducting a formal meta-analysis was beyond the scope of this review due to the heterogeneity of methodologies and outcome measures. However, we will undertake a cautious cross-study evaluation, reinterpreting the results with a focus on the variability in study designs and the potential for bias. We will include a revised discussion that emphasizes the descriptive nature of our analysis and the need for further research to establish correlations.
Comments 5: Accordingly, Figure 2 is not very meaningful either – it merely shows the variety of measurement methods that have led to completely different values.
Response 5: We understand that Figure 2, in its current form, lacks interpretative value. To enhance its utility, we will revise the figure to include key details (e.g., body height and body mass), allowing for a clearer comparison of variations in impact strength relative to anthropometric characteristics.
Comments 6: If a meta-analysis is not possible, at least a cross-study evaluation should be carried out and the results interpreted cautiously.
Response 6: We will revise the manuscript to provide a more cautious interpretation of the findings. The revised discussion will explicitly address the limitations of using mean values and highlight the need for standardized methodologies and comprehensive data collection in future research.
We are committed to addressing these concerns and will submit a revised manuscript that aligns with your valuable feedback. Thank you once again for your thoughtful review, which has greatly contributed to the improvement of our work.
Reviewer 3 Report
Comments and Suggestions for Authors
General comments
This manuscript aims to examine the influence of anthropometric characteristics, such as height and body mass, on the impact of punches in striking combat sports. The authors’ aim is commendable. The authors found that body mass is not the only determining factor, whereas higher weight categories are associated with greater punch impact. The relationship between height and impact also showed mixed results, with no clear association. The review highlights the lack of a gold standard instrument for evaluating punch impact. I do believe the authors guiltily neglected the issue of sport categorisation. Overall, the authors fulfil their aim sufficiently.
Specific comment
The manuscript completely neglects the issue of sport categorization.
Relevant missing ref:
https://pubmed.ncbi.nlm.nih.gov/32182165/
Minor comments
(line 14) Please, introduce PowerKube;
(l45) … (1,10-15)… Filimonov et al. (11) stated… (<- both, similarly, elsewhere throughout the manuscript, as well).
Author Response
Thank you very much for taking the time to review this manuscript. Please find the detailed responses below.
Comments 1: This manuscript aims to examine the influence of anthropometric characteristics, such as height and body mass, on the impact of punches in striking combat sports. The authors’ aim is commendable. The authors found that body mass is not the only determining factor, whereas higher weight categories are associated with greater punch impact. The relationship between height and impact also showed mixed results, with no clear association. The review highlights the lack of a gold standard instrument for evaluating punch impact. I do believe the authors guiltily neglected the issue of sport categorisation. Overall, the authors fulfil their aim sufficiently. The manuscript completely neglects the issue of sport categorization. Relevant missing ref: https://pubmed.ncbi.nlm.nih.gov/32182165/
Response 1: We acknowledge the editor's concern regarding the neglect of sport categorization. We understand that the type of combat sport may influence the impact of punches, as different sports emphasize various striking techniques and physical attributes. Therefore, we have revised the manuscript to explicitly address the role of sport categorization. We have also referenced the relevant study (https://pubmed.ncbi.nlm.nih.gov/32182165/) as suggested by the editor to strengthen this discussion.
Comments 2: (line 14) Please, introduce PowerKube;
Response 2: Line 14: We have introduced PowerKube as requested and included a brief description of the device in the revised manuscript. This clarification should provide readers with a better understanding of the tools used in the studies reviewed.
Comments 3: (l45) … (1,10-15)… Filimonov et al. (11) stated… (<- both, similarly, elsewhere throughout the manuscript, as well).
Response 3: Line 45: We have corrected the citation formatting as suggested. The references now follow the correct format throughout the manuscript. Specifically, we have amended the citation in line 45 to read: "Filimonov et al. (11) stated..." and applied the same correction throughout the manuscript wherever similar issues were found.
Thank you for your thoughtful comments, and we believe that the revisions will improve the clarity and comprehensiveness of the manuscript.
Round 2
Reviewer 3 Report
Comments and Suggestions for Authors
General comments
I do not have any further concerns about this manuscript. The authors addressed all the issues I raised well enough.